# DEMYSTIFYING LOSS FUNCTIONS FOR CLASSIFICATION

## ABSTRACT

It is common to use the softmax cross-entropy loss to train neural networks on classification datasets where a single class label is assigned to each example. However, it has been shown that modifying softmax cross-entropy with label smoothing or regularizers such as dropout can lead to higher performance. In this paper, we compare a variety of loss functions and output layer regularization strategies that improve performance on image classification tasks. We find differences in the outputs of networks trained with these different objectives, in terms of accuracy, calibration, out-of-distribution robustness, and predictions. However, differences in hidden representations of networks trained with different objectives are restricted to the last few layers; representational similarity reveals no differences among network layers that are not close to the output. We show that all objectives that improve over vanilla softmax loss produce greater class separation in the penultimate layer of the network, which potentially accounts for improved performance on the original task, but results in features that transfer worse to other tasks.

## 1 INTRODUCTION

Softmax cross-entropy (Bridle, 1990a;b) is the canonical loss function for multi-class classification in deep learning. However, the popularity of softmax cross-entropy appears to be driven by the aesthetic appeal of its probabilistic interpretation, rather than by practical superiority. Early studies reported no empirical advantage of softmax cross-entropy over squared-error loss (Richard & Lippmann, 1991; Weigend, 1993; Dietterich & Bakiri, 1994), and more recent work has found other objectives that yield better performance on certain tasks (e.g. Szegedy et al., 2016; Liu et al., 2016; Beyer et al., 2020). These studies show that it is possible to achieve meaningful improvements in accuracy simply by changing the loss function. Nonetheless, there has been little comparison among these alternative objectives, and even less investigation of *why* some objectives work better than others.

In this paper, we perform a comprehensive empirical study of the properties of 9 common and less-common loss functions and regularizers for deep learning, on standard image classification benchmarks. Most existing work in this area has proposed a new loss function or regularizer and attempted to demonstrate its superiority over a limited set of alternatives on benchmark tasks. This approach creates strong incentives to demonstrate the superiority of the proposed loss and little incentive to understand its limitations. Our goal is instead to understand when one might want to use one loss function or regularizer over another and, more broadly, to understand the extent to which neural network performance and representations can be manipulated through the choice of objective alone. Our key contributions are as follows:

- We rigorously benchmark 9 training objectives on standard image classification tasks, measuring accuracy, calibration, and out-of-distribution robustness. Many objectives improve over vanilla softmax cross-entropy loss, but no single objective performs best on all benchmarks.
- We demonstrate that different loss functions and regularizers produce different patterns of predictions, but combining them does not appear to improve accuracy. However, regularization that affects the input, such as AutoAugment (Cubuk et al., 2019) and Mixup (Zhang et al., 2017), can provide further gains. Our best models achieve state-of-the-art accuracy (79.1%/94.5% top-1/top-5) on ImageNet for unmodified ResNet-50 architectures trained from scratch.
- Using centered kernel alignment (CKA), we measure the similarity of the hidden representations of networks trained with different objectives. We show that the choice of objective affects representations in network layers close to the output, but earlier layers are highly similar regardless of what loss function is used.

- We show that all objectives that improve accuracy over softmax cross-entropy also lead to greater separation between representations of different classes in the penultimate layer. This improvement in class separation may be related to the boost in accuracy these objectives provide. However, representations with greater class separation are also more heavily specialized for the original task, and linear classifiers operating on these features perform substantially worse on transfer tasks.

## 2  LOSS FUNCTIONS AND OUTPUT LAYER REGULARIZERS

We investigate 9 loss functions and output layer regularizers. Let $\boldsymbol{\ell} \in \mathbb{R}^K$ denote the network's output ("logit") vector, and let $\boldsymbol{t} \in \{0, 1\}^K$ denote a one-hot vector of targets, where $\|\boldsymbol{t}\|_1 = 1$. Let $\boldsymbol{x} \in \mathbb{R}^M$ denote the vector of penultimate layer activations, which gives rise to the output vector as $\boldsymbol{\ell} = \boldsymbol{W}\boldsymbol{x} + \boldsymbol{b}$, where $\boldsymbol{W} \in \mathbb{R}^{K \times M}$ is the matrix of final layer weights, and $\boldsymbol{b}$ is a vector of biases.

All investigated loss functions include a term that encourages $\boldsymbol{\ell}$ to have a high dot product with $\boldsymbol{t}$. To avoid solutions that make this dot product large simply by increasing the scale of $\boldsymbol{\ell}$, these loss functions must also include one or more contractive terms and/or normalize $\boldsymbol{\ell}$. Many "regularizers" correspond to additional contractive terms added to the loss, so we do not draw a firm distinction between losses and regularizers. We describe each loss in detail below. Hyperparameters are provided in Appendix A.1.

**Softmax cross-entropy** (Bridle, 1990a;b) is the de facto loss function for multi-class classification in deep learning. It can be written as:

$$\mathcal{L}_{\text{softmax}}(\boldsymbol{\ell}, \boldsymbol{t}) = -\sum_{k=1}^{K} t_k \log\left(\frac{e^{\ell_k}}{\sum_{j=1}^{K} e^{\ell_j}}\right) = -\sum_{k=1}^{K} t_k \ell_k + \log\sum_{k=1}^{K} e^{\ell_k}. \tag{1}$$

The loss consists of a term that maximizes the dot product between the logits and targets, as well as a contractive term that minimizes the LogSumExp of the logits.

**Label smoothing** (Szegedy et al., 2016) "smooths" the targets for softmax cross-entropy loss. The new targets are given by mixing the original targets with a uniform distribution over all labels, $t' = t \times (1 - \alpha) + \alpha/K$, where $\alpha$ determines the weighting of the original and uniform targets. In order to maintain the same scale for the gradient with respect to the positive logit, in our experiments, we scale the label smoothing loss by $1/(1 - \alpha)$. The resulting loss is:

$$\mathcal{L}_{\text{smooth}}(\boldsymbol{\ell}, \boldsymbol{t}; \alpha) = -\frac{1}{1-\alpha}\sum_{k=1}^{K}\left((1-\alpha)t_k + \frac{\alpha}{K}\right)\log\left(\frac{e^{\ell_k}}{\sum_{j=1}^{K} e^{\ell_j}}\right) \tag{2}$$

$$= -\sum_{k=1}^{K} t_k \ell_k + \frac{1}{1-\alpha}\log\sum_{k=1}^{K} e^{\ell_k} - \frac{\alpha}{(1-\alpha)K}\sum_{k=1}^{K}\ell_k. \tag{3}$$

Compared to softmax cross-entropy loss, label smoothing adds an additional term that encourages the logits to be positive. Müller et al. (2019) previously showed that label smoothing improves calibration and encourages class centroids to lie at the vertices of a regular simplex.

**Dropout** (Srivastava et al., 2014) is among the most prominent regularizers in the deep learning literature. We consider dropout applied to the penultimate layer of the neural network, i.e., when inputs to the final layer are randomly kept with some probability $\rho$. When employing dropout, we replace the penultimate layer activations $\boldsymbol{x}$ with $\tilde{\boldsymbol{x}} = \boldsymbol{x} \odot \boldsymbol{\xi}/\rho$ where $\xi_i \sim \text{Bernoulli}(\rho)$. Writing the dropped out logits as $\tilde{\boldsymbol{\ell}} = \boldsymbol{W}\tilde{\boldsymbol{x}} + \boldsymbol{b}$, the dropout loss is:

$$\mathcal{L}_{\text{dropout}}(\boldsymbol{W}, \boldsymbol{b}, \boldsymbol{x}, \boldsymbol{t}; p) = \mathbb{E}_{\boldsymbol{\xi}}\left[\mathcal{L}_{\text{softmax}}(\tilde{\boldsymbol{\ell}}, \boldsymbol{t})\right] \tag{4}$$

Dropout produces both *implicit* regularization, by introducing noise into the optimization process, and *explicit* regularization, by altering the representation that minimizes the loss (Wei et al., 2020). Wager et al. (2013) have previously derived a quadratic approximation to the explicit regularizer for logistic regression and other generalized linear models; this strategy can also be used to approximate the explicit regularization imposed by dropout on the penultimate layer of a neural network with softmax loss. However, we observe that penultimate layer dropout has similar effects to extra final layer $L^2$ regularization, suggesting that implicit regularization is the more important component.

**Extra final layer $L^2$ regularization**: It is common to place the same $L^2$ regularization on the final layer as elsewhere in the network. However, we find that applying greater $L^2$ regularization to the

final layer can improve performance. In architectures with batch normalization, adding additional $L^2$ regularization has no explicit regularizing effect if the learnable scale ($\gamma$) parameters that are unregularized, but it still exerts an implicit regularizing effect by altering optimization.

**Logit penalty**: Whereas label smoothing encourages logits not to be too negative, and dropout imposes a penalty on the logits that depends on the covariance of the weights, an alternative possibility is simply to explicitly constrain logits to be small in $L^2$ norm:

$$\mathcal{L}_{\text{logit\_penalty}}(\boldsymbol{\ell}, \boldsymbol{t}; \beta) = \mathcal{L}_{\text{softmax}}(\boldsymbol{\ell}, \boldsymbol{t}) + \beta\|\boldsymbol{\ell}\|^2. \tag{5}$$

**Logit normalization**: We consider the use of $L^2$ *normalization*, rather than regularization, of the logits. Because the entropy of the output of the softmax function depends on the scale of the logits, which is lost after normalization, we introduce an additional temperature parameter $\tau$ that controls the magnitude of the logit vector, and thus, indirectly, the minimum entropy of the output distribution:

$$\mathcal{L}_{\text{logit\_norm}}(\boldsymbol{\ell}, \boldsymbol{t}; \tau) = \mathcal{L}_{\text{softmax}}(\boldsymbol{\ell}/(\tau\|\boldsymbol{\ell}\|), \boldsymbol{t}) \tag{6}$$

**Cosine softmax**: We additionally consider $L^2$ normalization of both the penultimate layer features and the final layer weights corresponding to each class. This loss is equivalent to softmax cross-entropy loss if the logits are given by cosine similarity $\text{sim}(\boldsymbol{x}, \boldsymbol{y}) = \boldsymbol{x}^\mathsf{T}\boldsymbol{y}/(\|\boldsymbol{x}\|\|\boldsymbol{y}\|)$ between the weight vector and the penultimate layer plus a per-class bias:

$$\mathcal{L}_{\text{cos\_softmax}}(\boldsymbol{W}, \boldsymbol{b}, \boldsymbol{x}, \boldsymbol{t}; \tau) = -\sum_{k=1}^{K} t_k \left(\text{sim}(\boldsymbol{W}_{k,:}, \boldsymbol{x})/\tau + b_k\right) + \log\sum_{k=1}^{K} e^{\text{sim}(\boldsymbol{W}_{k,:}, \boldsymbol{x})/\tau + b_k} \tag{7}$$

where $\tau$ is a temperature parameter as above. Similar losses have appeared in previous literature (Ranjan et al., 2017; Wojke & Bewley, 2018; Wang et al., 2018a;b; Deng et al., 2019; Liu et al., 2017), and variants have introduced explicit additive or multiplicative margins to this loss that we do not consider here (Liu et al., 2017; Wang et al., 2018a;b; Deng et al., 2019). It is possible that performance could be enhanced by employing one of these margin schemes, although we observe that manipulating the temperature alone has a large impact on observed class separation.

**Sigmoid cross-entropy** is the natural analog to softmax cross-entropy for multi-label classification problems. Although we investigate only single-label multi-class classification tasks, we train networks with sigmoid cross-entropy and evaluate accuracy by ranking the logits of the sigmoids. This approach is related to the one-versus-rest strategy for converting binary classifiers to multi-class classifiers. The sigmoid cross-entropy loss is:

$$\mathcal{L}_{\text{sigmoid}}(\boldsymbol{\ell}, \boldsymbol{t}) = -\sum_{k=1}^{K}\left(t_k \log\left(\frac{e^{\ell_k}}{e^{\ell_k}+1}\right) + (1-t_k)\log\left(1 - \frac{e^{\ell_k}}{e^{\ell_k}+1}\right)\right) \tag{8}$$

$$= -\sum_{k=1}^{K} t_k\ell_k + \sum_{k=1}^{K}\log(e^{\ell_k}+1). \tag{9}$$

The LogSumExp term of softmax loss is replaced with the sum of the softplus-transformed logits. We initialize the biases of the logits $\boldsymbol{b}$ to $-\log(K)$ so that the initial output probabilities are approximately $1/K$. Beyer et al. (2020) have previously shown that sigmoid cross-entropy loss leads to improved accuracy on ImageNet relative to softmax cross-entropy.

**Squared error**: Finally, we investigate squared error loss, as formulated by Hui & Belkin (2020):

$$\mathcal{L}_{\text{squared\_error}}(\boldsymbol{\ell}, \boldsymbol{t}; \kappa, M) = \frac{1}{K}\sum_{k=1}^{K}\left(\kappa t_k(\ell_k - M)^2 + (1-t_k)\ell_k^2\right) \tag{10}$$

where $\kappa$ and $M$ are hyperparameters. $\kappa$ sets the strength of the loss for the correct class relative to incorrect classes, whereas $M$ controls the magnitude of the correct class target. When $\kappa = M = 1$, the loss is simply the mean squared error between $\boldsymbol{\ell}$ and $\boldsymbol{t}$. Like Hui & Belkin (2020), we find that placing greater weight on the correct class slightly improves ImageNet accuracy.

## 3 RESULTS

For each loss, we trained 8 ResNet-50 (He et al., 2016; Gross & Wilber, 2016) models on ImageNet. To tune loss hyperparameters and the epoch for early stopping, we performed 3 training runs per hyperparameter configuration where we held out a validation set of 50,046 ImageNet training example.

**Table 1: Regularizers and alternative losses improve ImageNet accuracy.** Accuracy of models trained with different losses/regularizers on the ImageNet validation (mean $\pm$ standard error of 8 models) and CIFAR-10 and CIFAR-100 test sets (mean $\pm$ standard error of 25 models). Losses are sorted from lowest to highest ImageNet top-1 accuracy. Accuracy values not significantly different from the best ($p > 0.05$, t-test) are bold-faced.

| Loss/regularizer | ImageNet (ResNet-50) | | CIFAR-10 | CIFAR-100 |
| | Top-1 Acc. (%) | Top-5 Acc. (%) | (All-CNN-C + BN) | (WRN 16-8) |
|---|---|---|---|---|
| Softmax | $77.0 \pm 0.06$ | $93.40 \pm 0.02$ | $93.49 \pm 0.03$ | $79.7 \pm 0.04$ |
| Squared error | $77.2 \pm 0.04$ | $92.79 \pm 0.02$ | $93.31 \pm 0.02$ | $79.4 \pm 0.05$ |
| Dropout | $77.5 \pm 0.04$ | $93.62 \pm 0.02$ | $93.74 \pm 0.03$ | $79.5 \pm 0.06$ |
| Label smoothing | $77.6 \pm 0.03$ | $93.78 \pm 0.01$ | $\mathbf{93.79} \pm 0.03$ | $80.0 \pm 0.05$ |
| Extra final layer $L^2$ | $77.7 \pm 0.03$ | $93.79 \pm 0.02$ | $93.63 \pm 0.03$ | $\mathbf{80.2} \pm 0.05$ |
| Logit penalty | $77.7 \pm 0.02$ | $\mathbf{93.83} \pm 0.02$ | $\mathbf{93.84} \pm 0.04$ | $\mathbf{80.2} \pm 0.05$ |
| Logit normalization | $\mathbf{77.8} \pm 0.02$ | $93.71 \pm 0.02$ | $93.55 \pm 0.03$ | $78.9 \pm 0.05$ |
| Cosine softmax | $\mathbf{77.9} \pm 0.02$ | $\mathbf{93.86} \pm 0.01$ | $93.64 \pm 0.03$ | $\mathbf{80.1} \pm 0.06$ |
| Sigmoid | $\mathbf{77.9} \pm 0.05$ | $93.50 \pm 0.02$ | $\mathbf{93.79} \pm 0.04$ | $80.0 \pm 0.05$ |

We also trained 25 batch-normalized All-CNN-C (Springenberg et al., 2014) models for each loss on CIFAR-10 (Krizhevsky & Hinton, 2009), where we performed extensive hyperparameter tuning for learning rate and weight decay in addition to loss hyperparameters. We provide further details regarding training and hyperparameter selection in Appendix A.1.

## 3.1 REGULARIZERS AND ALTERNATIVE LOSSES ENHANCE ACCURACY

We found that, when properly tuned, many investigated objectives often provide a statistically significant improvement over softmax cross-entropy, as shown in Table 1. The range of improvements was small, but meaningful, with sigmoid cross-entropy and cosine softmax both leading to an improvement of 0.9% in top-1 accuracy over the baseline for ResNet-50 on ImageNet. No single loss performed best across all benchmarks, although cosine softmax, logit penalty, and sigmoid were frequently among the top-performing losses.

Losses that yielded large improvements in top-1 accuracy on ImageNet did not necessarily improve top-5 accuracy. For ResNet-50, sigmoid cross-entropy led to a large (0.9%) improvement in top-1 accuracy over vanilla softmax cross-entropy, but only a small (0.1%) improvement in top-5 accuracy. Cosine softmax performed comparably to sigmoid cross-entropy in terms of top-1 accuracy, but better in top-5 accuracy, with a 0.4% improvement over the baseline. Similar patterns were observed for Inception v3 (Table B.1), where sigmoid cross-entropy was the best-performing model in terms of top-1 accuracy but performed worse than the softmax baseline in terms of top-5 accuracy.

Losses also differed in out-of-distribution robustness, and in the calibration of the resulting predictions. Table B.2 shows results on the out-of-distribution test sets ImageNet-v2 (Recht et al., 2019), ImageNet-A (Hendrycks et al., 2019), ImageNet-Sketch (Wang et al., 2019), ImageNet-R (Hendrycks et al., 2020), and ImageNet-C (Hendrycks & Dietterich, 2019). In almost all cases, alternative loss functions outperformed softmax cross-entropy, with logit normalization and cosine softmax typically performing slightly better than alternatives. Effects on calibration, shown in Table B.3, were mixed. Label smoothing substantially reduced expected calibration error (Guo et al., 2017), as previously shown by Müller et al. (2019), although cosine softmax achieved a lower negative log likelihood. However, there was no clear relationship between calibration and accuracy. Although logit penalty performed well in terms of accuracy, it provided the worst calibration of any objective investigated.

Our attempts to achieve higher accuracy by combining objectives were unsuccessful. As described in Appendix C, adding additional regularization did not improve performance of well-tuned loss functions, and normalized variants of sigmoid cross-entropy loss failed to improve accuracy on ImageNet. However, it was still possible to improve networks' performance substantially using AutoAugment (Cubuk et al., 2019) or Mixup (Zhang et al., 2017), and gains from improved losses and these data augmentation strategies were approximately additive (Table C.2). With longer training, both sigmoid cross-entropy and cosine softmax achieve state-of-the-art accuracy among ResNet-50 networks trained with AutoAugment (Table C.3), matching or outperforming supervised contrastive learning (Khosla et al., 2020). Combining cosine softmax loss, AutoAugment, and Mixup, we achieve 79.1% top-1 accuracy and 94.5% top-5 accuracy, which is to our knowledge the best reported $224 \times 224$ pixel single-crop accuracy with an unmodified ResNet-50 architecture trained from scratch.

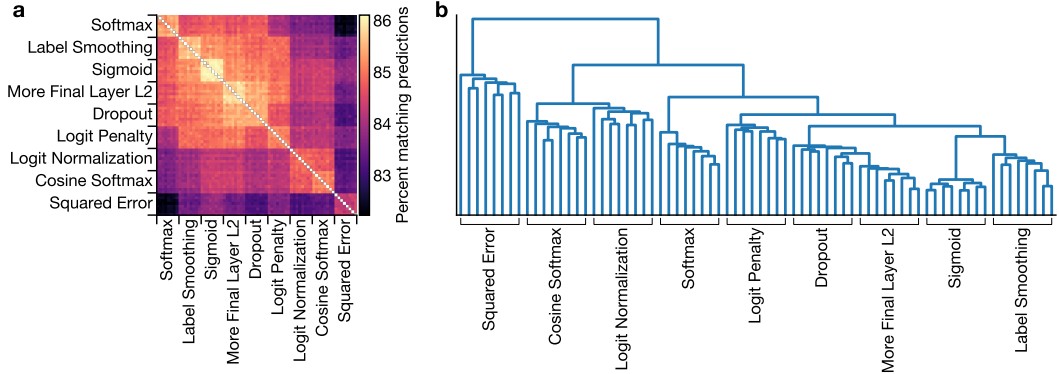

**Figure 1: Different losses produce different predictions. a**: Percentages of ImageNet validation set examples for which models assign the same top-1 predictions, for 8 seeds of ResNet-50 models. **b**: Dendrogram based on similarity of predictions. All models naturally cluster according to loss, except for "Dropout" and "More Final Layer L2" models. See also Figure D.1.

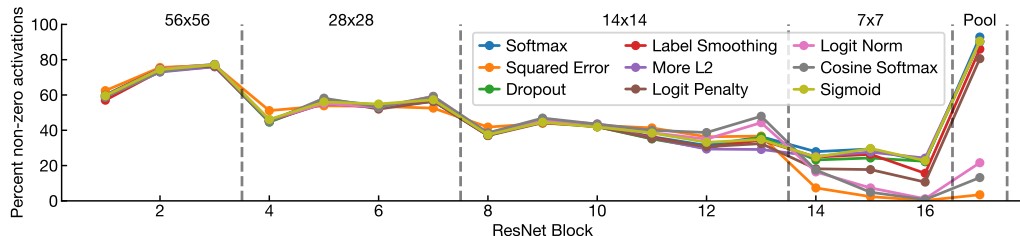

**Figure 2: Loss functions affect sparsity of later layer representations.** Plot shows the average % non-zero activations for each ResNet-50 block, after the residual connection and subsequent nonlinearity, on the ImageNet validation set. Dashed lines indicate boundaries between stages.

### 3.2 DIFFERENT LOSSES PRODUCE DIFFERENT PREDICTIONS

Given that effects of regularization were non-additive, we sought to determine whether different regularizers and losses had similar effects on network predictions. For each pair of models, we measured the percentage of images in the ImageNet validation set where both models predicted the same class. The results are shown in Figure 1. We also examined the percentage of images that where both models are either correct or incorrect, and the agreement on examples that both models get incorrect (Figure D.1). All ways of measuring similarity of predictions yielded similar results.

Models' predictions clustered into distinct groups according to their loss functions. Models trained from different initializations with the same loss function were more similar than models trained with different loss functions. However, all models trained with (regularized) softmax loss or sigmoid loss were more similar to each other than they were to models trained with logit or feature + weight normalization. Networks trained with squared error were dissimilar to all others examined.

Variability in predictions of models trained with the same loss but different random initializations was large. Although standard deviations in top-1 accuracy were <0.2% for all losses, even the most similar pair of models disagreed on 13.9% of test set examples. When ensembling the 8 models trained with the same loss but different random initializations, the least similar losses (softmax and squared error) disagreed on only 11.5% of examples (Figure D.2). The accuracy of ensembles of models trained with different losses was closely related to the accuracies of the constituent models; ensembling models trained with the two best losses yielded only modest accuracy improvements over ensembles trained with either loss alone (Figure D.3).

### 3.3 LOSSES PRIMARILY AFFECT HIDDEN REPRESENTATIONS CLOSE TO THE OUTPUT

Loss functions differ not only in their predictions, but also in their effects on internal representations of neural networks. In Figure 2, we show the sparsity of the activations of layers of networks trained with different loss functions. In all networks, the percentage of non-zero ReLU activations decreased with depth, attaining its minimum at the last convolutional layer. In the first three ResNet stages, activation sparsity was broadly similar regardless of the loss. However, in the final stage and penultimate average pooling layer, there were substantial differences.

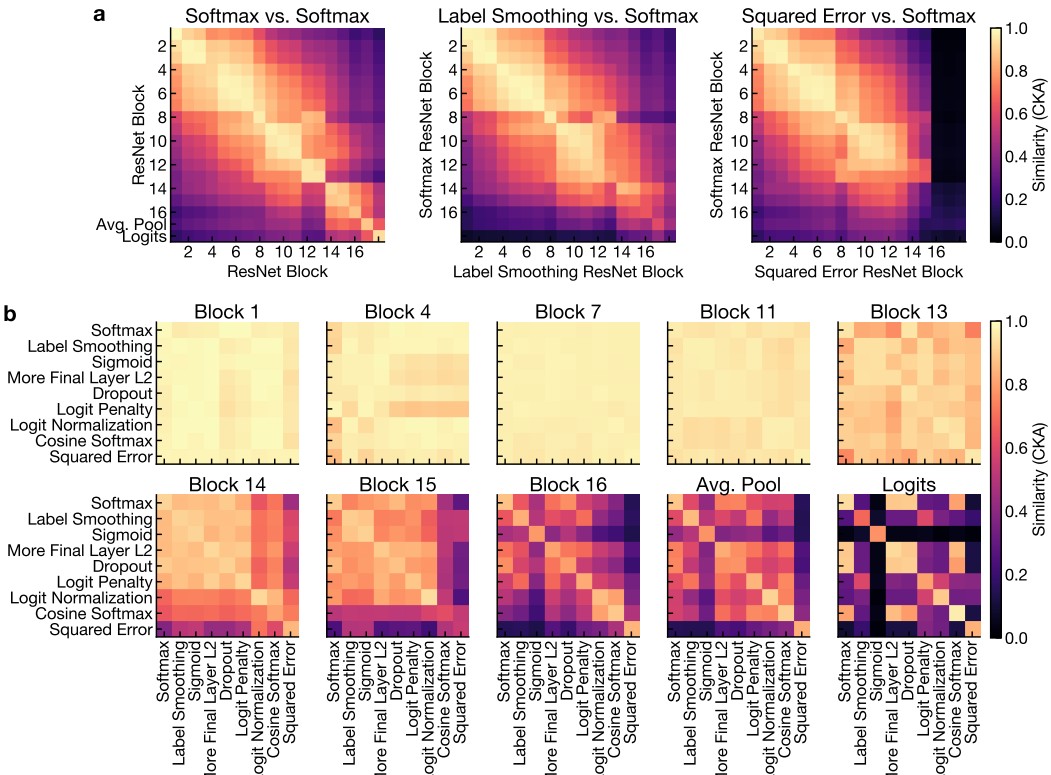

**Figure 3: The loss function has little impact on representations in early network layers.** All plots show linear centered kernel alignment (CKA) between representations computed on the ImageNet validation set. **a**: CKA between network layers, for pairs of networks trained from different initializations. **b**: CKA between representations extracted from architecturally corresponding layers of networks trained with different loss functions. Diagonal reflects similarity of networks with the same loss function trained from different initalizations.

Given that these observations, we wondered whether the choice of loss had any effect on representations in these layers at all. We used linear centered kernel alignment (CKA) (Kornblith et al., 2019a; Cortes et al., 2012; Cristianini et al., 2002) to measure the similarity between networks' hidden representations. As shown in Figure 3, representations of corresponding early, but not late, network layers were highly similar regardless of loss function. These results provide further confirmation that effects of the loss function are limited to later network layers.

### 3.4 REGULARIZATION IMPROVES CLASS SEPARATION

Is there a feature of the investigated regularizers that can potentially explain their beneficial effect on accuracy? We demonstrate that all investigated regularizers and alternative losses force the network to shrink or eliminate directions in the penultimate layer representation space that are not aligned with weight vectors. The universality of this finding suggests it may relate to the accuracy-enhancing properties of these losses.

The ratio of the average within-class cosine distance to the overall average cosine distance provides a measure of how distributed examples within a class are that is between 0 and 1. We take one minus this quantity to get a closed-form measure of class separation:

$$R^2 = 1 - \frac{\sum_{k=1}^{K} \sum_{m=1}^{N_k} \sum_{n=1}^{N_k} \left(1 - \text{sim}(\boldsymbol{x}_{k,m}, \boldsymbol{x}_{k,n})\right)/N_K^2}{\sum_{j=1}^{K} \sum_{k=1}^{K} \sum_{m=1}^{N_j} \sum_{n=1}^{N_k} \left(1 - \text{sim}(\boldsymbol{x}_{j,m}, \boldsymbol{x}_{k,n})\right)/(N_j N_k)} \tag{11}$$

where $\boldsymbol{x}_{k,m}$ is the embedding of example $m$ in class $k$, $N_k$ is the number of examples in class $k$, and $\text{sim}(\boldsymbol{x}, \boldsymbol{y}) = \boldsymbol{x}^\mathsf{T} \boldsymbol{y}/(\|\boldsymbol{x}\|\|\boldsymbol{y}\|)$ is cosine similarity between vectors. If the embeddings are first $L^2$ normalized, then $1 - R^2$ is the ratio of the average within-class variance to the weighted total variance, where the weights are inversely proportional to the number of examples in each class. For a balanced dataset, $R^2$ is also equivalent to centered kernel alignment (Cortes et al., 2012; Cristianini et al., 2002) between the embeddings and the one-hot label matrix, with a cosine kernel. We also examined alternative class separation metrics (Appendix E); results were similar.

**Table 2: Regularization and alternative losses improve class separation in the penultimate layer.** Results averaged over 8 ResNet-50 models per loss on the Image-Net training set.

| Loss/regularizer | Class separation ($R^2$) |
|---|---|
| Softmax | $0.3494 \pm 0.0002$ |
| Squared error | $0.8452 \pm 0.0002$ |
| Dropout | $0.4606 \pm 0.0003$ |
| Label smoothing | $0.4197 \pm 0.0003$ |
| Extra $L^2$ | $0.5718 \pm 0.0006$ |
| Logit penalty | $0.6012 \pm 0.0004$ |
| Logit norm | $0.5167 \pm 0.0002$ |
| Cosine softmax | $0.6406 \pm 0.0003$ |
| Sigmoid | $0.4267 \pm 0.0003$ |

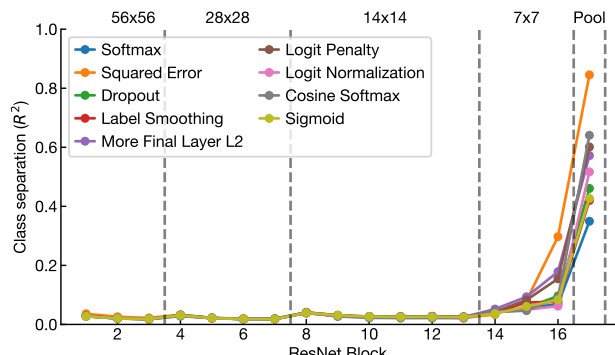

**Figure 4:** Class separation in different layers of ResNet-50 models, on the ImageNet training set.

**Table 3: Regularized networks learn features specialized to ImageNet.** Accuracy of linear classifiers ($L^2$-regularized multinomial logistic regression) trained to classify different datasets using fixed penultimate layer features. IN(50k) reflects accuracy of a classifier trained on 50,046 examples from the ImageNet training set and tested on the validation set. See Appendix A.2 for training details.

| Pretraining loss | Food | CIFAR10 | CIFAR100 | Birdsnap | SUN397 | Cars | Pets | Flowers | IN(50k) |
|---|---|---|---|---|---|---|---|---|---|
| Softmax | **74.6** | **92.4** | **76.9** | **55.4** | **62.0** | **60.3** | 92.0 | **94.0** | 71.1 |
| Squared error | 39.8 | 82.2 | 56.3 | 21.8 | 39.9 | 15.3 | 84.7 | 46.7 | 76.7 |
| Dropout | 72.6 | 91.4 | 75.0 | 53.6 | 61.2 | 54.7 | 92.6 | 92.1 | 74.8 |
| Label smoothing | 72.7 | 91.6 | 75.2 | 53.6 | 61.6 | 54.8 | **92.9** | 91.9 | 74.5 |
| Extra $L^2$ | 70.6 | 91.0 | 73.7 | 51.5 | 60.1 | 50.3 | 92.4 | 89.8 | 75.9 |
| Logit penalty | 68.1 | 90.2 | 72.3 | 48.1 | 59.0 | 48.3 | 92.3 | 86.6 | 76.4 |
| Logit norm | 66.3 | 90.5 | 72.9 | 50.7 | 58.1 | 45.4 | 92.0 | 82.9 | 75.1 |
| Cosine softmax | 62.0 | 89.9 | 71.3 | 45.4 | 55.0 | 36.7 | 91.1 | 75.3 | **76.9** |
| Sigmoid | 73.4 | 91.7 | 75.7 | 52.3 | **62.0** | 56.1 | 92.5 | 92.9 | 74.3 |

As shown in Table 2 and Figure 4, all regularizers and alternative loss functions resulted in greater class separation in penultimate (average pooling) layer representations as compared to softmax loss. Whereas additional final layer $L^2$, logit penalty, and squared error also produced greater class separation before the penultimate layer, other losses did not.

Although losses that improve class separation also improve accuracy on the ImageNet validation set, they result in penultimate layer features that are substantially less useful for other tasks. Kornblith et al. (2019b) previously showed that networks trained with label smoothing and dropout learn less transferable features. As in this work, we trained logistic regression classifiers to classify a selection of transfer datasets (Bossard et al., 2014; Krizhevsky & Hinton, 2009; Berg et al., 2014; Xiao et al., 2010; Krause et al., 2013; Parkhi et al., 2012; Nilsback & Zisserman, 2008), using fixed features from networks trained with different losses. As shown in Table 3, features from networks trained with vanilla softmax loss yield the highest transfer accuracy. However, when we attempted to relearn the original 1000-way ImageNet classifier using 50,046 training set examples, features from networks trained with vanilla softmax loss performed worst. Thus, the ease with which ImageNet classifier weights can be relearned from representations is inversely related to the performance of these representations when they are used to classify other datasets (Figure 5).

To confirm this relationship between class separation, ImageNet accuracy, and transfer, we trained models with cosine softmax with varying values of the temperature parameter $\tau$.[1] As shown in Table 4, lower temperatures resulted in lower top-1 accuracies and worse class separation, and made the ImageNet classifier weights more difficult to recover. However, even though the lowest temperature achieved 2.7% lower accuracy on ImageNet compared to higher temperatures, this lowest temperature yielded the better features for nearly all transfer datasets. Thus, $\tau$ controls a tradeoff between the generalizability of penultimate-layer features and the accuracy on the target dataset.

---

[1]Training at low temperatures was unstable, so we scaled the loss by the temperature, which slightly worsened overall ImageNet accuracy. Relationships for temperatures >= 0.05 remain consistent without loss scaling.

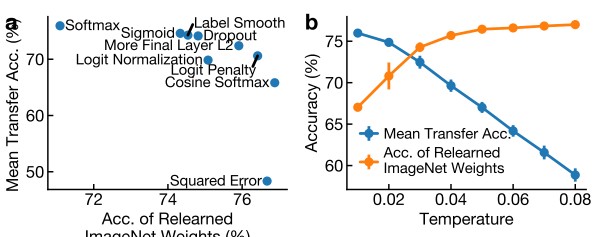

**Figure 5: Transfer accuracy and accuracy of relearned ImageNet weights are negatively related. a**: Average transfer task accuracy versus accuracy of a classifier trained on 50,046 ImageNet training set examples and tested on the validation set for different objectives. **b**: Relationship of transfer accuracy and relearned ImageNet accuracy with cosine softmax temperature.

**Table 4: Temperature of cosine softmax loss controls ImageNet top-1 accuracy, class separation ($R^2$), and linear transfer accuracy.**

| Temp. | ImageNet Top-1 | $R^2$ | Food | CIFAR10 | CIFAR100 | Birdsnap | SUN397 | Cars | Pets | Flowers | IN(50k) |
|---|---|---|---|---|---|---|---|---|---|---|---|
| 0.01 | 74.9 | 0.236 | **73.4** | **91.9** | **76.5** | **57.2** | **60.5** | **62.9** | 91.7 | **93.6** | 67.0 |
| 0.02 | 77.0 | 0.358 | 72.1 | 91.8 | 76.2 | 56.5 | 60.4 | 58.5 | 92.2 | 91.2 | 70.8 |
| 0.03 | 77.5 | 0.475 | 69.1 | 91.5 | 74.9 | 53.7 | 59.1 | 51.8 | **92.3** | 87.4 | 74.3 |
| 0.04 | **77.6** | 0.562 | 66.0 | 90.7 | 73.8 | 50.3 | 57.4 | 45.1 | 91.7 | 82.2 | 75.7 |
| 0.05 | **77.6** | 0.634 | 62.8 | 90.4 | 72.2 | 47.6 | 55.4 | 38.6 | 91.0 | 78.3 | 76.4 |
| 0.06 | 77.5 | 0.693 | 60.3 | 89.3 | 69.8 | 43.3 | 53.8 | 33.3 | 91.0 | 72.7 | 76.6 |
| 0.07 | 77.5 | 0.738 | 57.1 | 88.7 | 68.6 | 39.6 | 51.4 | 29.1 | 90.2 | 67.9 | 76.8 |
| 0.08 | **77.6** | **0.770** | 53.7 | 87.7 | 66.5 | 35.5 | 49.4 | 25.7 | 89.3 | 63.2 | **77.0** |

## 4 RELATED WORK

Theoretical analysis of loss functions is challenging; in most cases, solutions cannot be expressed in closed form even when the predictor is linear. However, Soudry et al. (2018) have previously shown that, on linearly separable data, gradient descent on the unregularized logistic or multinomial logistic regression objectives (i.e., linear models with sigmoid or softmax cross-entropy loss) eventually converges to the minimum norm solution. These results can be extended to neural networks in certain restricted settings (Soudry et al., 2018; Gunasekar et al., 2018; Wei et al., 2019).

Our study of class separation in penultimate layers of neural networks is related to work investigating angular visual hardness (Chen et al., 2019), which measures the arccosine-transformed cosine similarity between the weight vectors and examples. This metric is similar to the class separation metric we apply (Eq. 11), but fails to differentiate between networks trained with softmax and sigmoid cross-entropy; see Appendix Figure E.1. Other work has investigated how class information evolves through the hidden layers of neural networks, using linear classifiers (Alain & Bengio, 2016), binning estimators of mutual information (Shwartz-Ziv & Tishby, 2017; Saxe et al., 2019; Goldfeld et al., 2018), Euclidean distances (Schilling et al., 2018), and manifold geometry (Cohen et al., 2020). However, this previous work has not analyzed how training objectives affect these measures.

The loss functions we investigate are only a subset of those explored in past literature. We have excluded loss functions that require specially constructed batches from the current investigation (Snell et al., 2017; Khosla et al., 2020), as well as losses designed for situations with high label noise (Jindal et al., 2016; Ghosh et al., 2017; Patrini et al., 2017; Amid et al., 2019; Lukasik et al., 2020). Other work has investigated replacing the softmax function with other functions that lead to normalized class probabilities (de Brébisson & Vincent, 2015; Laha et al., 2018). Our approach is related to previous studies of metric learning (Musgrave et al., 2020) and optimizers (Choi et al., 2019).

## 5 CONCLUSION

Our study identifies many similarities among networks trained with different objectives. On CIFAR-10, CIFAR-100, and ImageNet, different losses and regularizers achieve broadly similar accuracies. Although the accuracy differences are large enough to be meaningful in some contexts, the largest is still <1.5%. Representational similarity analysis using centered kernel alignment indicates that the choice of loss function affects representations in only the last few layers of the network, suggesting inherent limitations to what can be achieved by manipulating the loss. However, we also show that different objectives lead to substantially different penultimate layer representations. We find that class separation is an important factor that distinguishes these different penultimate layer representations, and show that it is inversely related to transferability of representations to other tasks.

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

# Appendix

## A    DETAILS OF TRAINING AND HYPERPARAMETER TUNING

### A.1    TRAINING AND TUNING NEURAL NETWORKS

**ImageNet.**   We trained ImageNet models (ResNet-50 (He et al., 2016; Gross & Wilber, 2016; Goyal et al., 2017) "v1.5"[2] and Inception v3 (Szegedy et al., 2016)) models with SGD with Nesterov momentum of 0.9 and a batch size 4096 and weight decay of $8 \times 10^{-5}$ (applied to the weights but not batch norm parameters). After 10 epochs of linear warmup to a maximum learning rate of 1.6, we decayed the learning rate by a factor of 0.975 per epoch. We took an exponential moving average of the weights over training as in Szegedy et al. (2016), with a momentum factor of 0.9999. We used standard data augmentation comprising random crops of 10-100% of the image with aspect ratios of 0.75 to 1.33 and random horizontal flips. At test time, we resized images to 256 pixels on their shortest side and took a $224 \times 224$ center crop.

To tune hyperparameters, we initially performed a set of training runs with a wide range of different parameters, and then narrowed the hyperparameter range to the range shown in Table A.1. To further tune the hyperparameters and the epoch for early stopping, we performed 3 training runs per configuration where we held out a validation set of approximately 50,000 ImageNet training examples.[3] We tuned loss hyperparameters for ResNet-50 only. For Inception v3, we used the same loss hyperparameters as for ResNet-50, but we still performed 3 training runs with the held out validation set to select the point at which to stop for each loss.

**Table A.1: Hyperparameters for ImageNet.**

| Loss/regularizer | Hyperparameters | Epochs |
|---|---|---|
| Softmax | N/A | 146 |
| Squared error | $\kappa = 9$, $M = 60$, loss scale $= 10$ | 196 |
| Dropout | $\rho = \{0.6, 0.65, \mathbf{0.7}, 0.75, 0.8, 0.85\}$ | 172 |
| Label smoothing | $\alpha = \{0.08, 0.09, \mathbf{0.1}, 0.11.0.12\}$ | 180 |
| Extra final layer $L^2$ | $\lambda_{\text{final}} = \{4\text{e-}4, 6\text{e-}4, \mathbf{8\text{e-}4}, 1\text{e-}3\}$ | 168 |
| Logit penalty | $\beta = \{5\text{e-}5, 1\text{e-}4, 2\text{e-}4, 4\text{e-}4, \mathbf{6\text{e-}4}, 8\text{e-}4\}$ | 180 |
| Logit normalization | $\tau = \{0.03, \mathbf{0.04}, 0.05, 0.06\}$ | 152 |
| Cosine softmax | $\tau = \{0.04, 0.045, \mathbf{0.05}, 0.06, 0.07, 0.08\}$ | 158 |
| Sigmoid | N/A | 166 |

**CIFAR.**   We trained CIFAR-10 and CIFAR-100 models using SGD with Nesterov momentum of 0.9 and a cosine learning rate decay schedule without restarts, and without weight averaging. For CIFAR-10, we used a batch size of 128; for CIFAR-100, we used a batch size of 256. For these networks, we performed hyperparameter tuning to select the learning rate and weight decay parameters. We started by selecting the learning rate from $\{10^{-2}, 10^{-1.5}, 10^{-1}, 10^{-0.5}, 1.0, 10^{0.5}\}$ and the weight decay from $\{10^{-4.5}, 10^{-4}, 10^{-3.5}, 10^{-3}\}$, where we parameterize the weight decay so that it is divided by the learning rate. We manually inspected hyperparameter grids and expanded the learning rate and weight decay ranges when the best accuracy was on the edge of the searched grid. After finding the best hyperparameters in this coarse search, we performed a finer search in the vicinity of the best coarse hyperparameters with double the granularity, e.g., for an optimal learning rate of $10^{-1}$ in the coarse search, our fine grid would include learning rates of $\{10^{-0.5}, 10^{-0.75}, 10^{-1}, 10^{-1.25}, 10^{-1.5}\}$. All results show optimal hyperparameters from this finer

---

[2]The `torchvision` ResNet-50 model and the "official" TensorFlow ResNet both implement this architecture, which was first proposed by Gross & Wilber (2016) and differs from the ResNet v1 described by He et al. (2016) in performing strided convolution in the first $3 \times 3$ convolution in each stage rather than the first $1 \times 1$ convolution. Our implementation initializes the $\gamma$ parameters of the last batch normalization layer in each block to 0, as in Goyal et al. (2017).

[3]Due to the large number of hyperparameter configurations, for squared error, we performed only 1 run per configuration to select hyperparameters, but 3 to select the epoch at which to stop. We manually narrowed the hyperparameter search range until all trained networks achieved similar accuracy. The resulting hyperparaameters performed better than those suggested by Hui & Belkin (2020).

grid. During both coarse and fine hyperparameter tuning, we computed accuracies averaged over 5 different initializations for each configuration to reduce the bias toward selecting high-variance hyperparameter combinations when searching over a large number of configurations. Hyperparameters are shown in Table A.2.

The architecture we used for CIFAR-10 experiments was based on All-CNN-C architecture of Springenberg et al. (2014), with batch normalization added between layers and the global average pooling operation moved before the final convolutional layer. On CIFAR-100, we used the Wide ResNet 16-8 architecture from Zagoruyko & Komodakis (2016). Our CIFAR-100 architecture applied weight decay to batch normalization parameters, but our CIFAR-10 architecture did not.

**Table A.2: Hyperparameters for CIFAR.** $\eta$ is the learning rate and $\tilde{\lambda}$ is the *product* of the learning rate and the weight decay added to the loss, i.e., the weight decay loss is $\mathcal{L}_{\text{weight\_decay}} = \frac{\tilde{\lambda}}{2\eta}\|\mathbf{w}\|^2$.

| Loss/regularizer | CIFAR-10 (All-CNN-C + BN) | CIFAR-100 (WRN 16-8) |
|---|---|---|
| Squared error | $\eta = 0.1, \tilde{\lambda} = 10^{-3.5}, \kappa = 8, M = 0.83$ | $\eta = 0.1, \tilde{\lambda} = 10^{-3.75}, \kappa = 6, M = 12$ |
| Softmax | $\eta = 10^{-0.75}, \tilde{\lambda} = 10^{-3.75}$ | $\eta = 0.1, \tilde{\lambda} = 10^{-4}$ |
| Logit normalization | $\eta = 0.01, \tilde{\lambda} = 10^{-4}, \tau = 0.14$ | $\eta = 10^{-2.25}, \tilde{\lambda} = 10^{-3.75}, \tau = 0.11$ |
| Extra final layer $L^2$ | $\eta = 0.1, \tilde{\lambda} = 10^{-3.5}, \lambda_{\text{final}} = 10^{-1.5}$ | $\eta = 0.1, \tilde{\lambda} = 10^{-3.75}, \lambda_{\text{final}} = 10^{-3.33}$ |
| Cosine softmax | $\eta = 10^{-2.25}, \tilde{\lambda} = 10^{-4}, \tau = 0.08$ | $\eta = 0.01, \tilde{\lambda} = 10^{-3.75}, \tau = 0.1$ |
| Dropout | $\eta = 0.1, \tilde{\lambda} = 10^{-3.75}, \rho = 0.65$ | $\eta = 10^{-1.25}, \tilde{\lambda} = 10^{-3.75}, \rho = 0.75$ |
| Sigmoid | $\eta = 1, \tilde{\lambda} = 10^{-3.75}$ | $\eta = 0.1, \tilde{\lambda} = 10^{-3.75}$ |
| Label smoothing | $\eta = 0.1, \tilde{\lambda} = 10^{-3.75}, \alpha = 0.04$ | $\eta = 0.1, \tilde{\lambda} = 10^{-3.5}, \alpha = 0.18$ |
| Logit penalty | $\eta = 10^{-0.75}, \tilde{\lambda} = 10^{-3.75}, \beta = 10^{-2.83}$ | $\eta = 10^{-1.25}, \tilde{\lambda} = 10^{-3.75}, \beta = 10^{-2.83}$ |

## A.2 Training and Tuning Multinomial Logistic Regression Classifiers

To train multinomial logistic regression classifiers on fixed features, we follow a similar approach to Kornblith et al. (2019b). We first extracted features for every image in the training set, by resizing them to 224 pixels on the shortest side and taking a $224 \times 224$ pixel center crop. We held out a validation set from the training set, and used this validation set to select the $L^2$ regularization hyperparameter, which we selected from 45 logarithmically spaced values between $10^{-6}$ and $10^5$, applied to the sum of the per-example losses. Because the optimization problem is convex, we used the previous weights as a warm start as we increased the $L^2$ regularization hyperparameter. After finding the optimal hyperparameter on this validation set, we retrained on the entire training set and evaluated accuracy on the test set.

## B Additional Evaluation of Performance of Regularizers and Losses

**Table B.1: Regularizers and alternative losses improve Inception v3 accuracy on ImageNet.** Accuracy (mean $\pm$ standard error of 3 models) with different losses/regularizers on the ImageNet validation set. Losses are sorted from lowest to highest top-1 accuracy. Accuracy values not significantly different from the best ($p > 0.05$, t-test) are bold-faced.

| Loss/regularizer | Top-1 Acc. (%) | Top-5 Acc. (%) |
|---|---|---|
| Squared error | $77.7 \pm 0.03$ | $93.28 \pm 0.01$ |
| Softmax | $78.6 \pm 0.03$ | $94.24 \pm 0.03$ |
| Logit normalization | $\mathbf{78.8} \pm 0.11$ | $94.34 \pm 0.04$ |
| Label smoothing | $78.8 \pm 0.03$ | $\mathbf{94.60} \pm 0.03$ |
| Cosine softmax | $\mathbf{78.9} \pm 0.06$ | $94.38 \pm 0.03$ |
| Logit penalty | $\mathbf{78.9} \pm 0.06$ | $\mathbf{94.63} \pm 0.02$ |
| Dropout | $\mathbf{79.0} \pm 0.02$ | $\mathbf{94.50} \pm 0.04$ |
| Extra final layer $L^2$ | $\mathbf{79.0} \pm 0.03$ | $94.52 \pm 0.01$ |
| Sigmoid | $\mathbf{79.1} \pm 0.07$ | $94.17 \pm 0.02$ |

**Table B.2: Regularizers and alternative losses improve performance on out-of-distribution test sets.** Accuracy averaged over 8 ResNet-50 models per loss.

| Loss/regularizer | ImageNet-v2 (%) | ImageNet-A (%) | IN-Sketch (%) | ImageNet-R (%) | ImageNet-C (mCE) |
|---|---|---|---|---|---|
| Softmax | $65.0 \pm 0.1$ | $2.7 \pm 0.0$ | $21.8 \pm 0.1$ | $36.8 \pm 0.1$ | $75.9 \pm 0.1$ |
| Squared error | $65.3 \pm 0.1$ | $4.5 \pm 0.1$ | $22.4 \pm 0.1$ | $36.3 \pm 0.1$ | $74.6 \pm 0.1$ |
| Dropout | $65.4 \pm 0.0$ | $3.1 \pm 0.1$ | $23.0 \pm 0.1$ | $37.2 \pm 0.1$ | $74.5 \pm 0.1$ |
| Label smoothing | $\mathbf{65.7} \pm 0.1$ | $3.8 \pm 0.1$ | $22.5 \pm 0.1$ | $37.8 \pm 0.1$ | $75.2 \pm 0.1$ |
| Extra final layer $L^2$ | $\mathbf{65.8} \pm 0.1$ | $3.3 \pm 0.0$ | $23.1 \pm 0.1$ | $37.7 \pm 0.1$ | $74.1 \pm 0.1$ |
| Logit penalty | $\mathbf{65.8} \pm 0.0$ | $4.5 \pm 0.0$ | $22.8 \pm 0.1$ | $38.1 \pm 0.1$ | $74.3 \pm 0.1$ |
| Logit normalization | $\mathbf{65.8} \pm 0.1$ | $\mathbf{4.8} \pm 0.1$ | $23.7 \pm 0.1$ | $\mathbf{39.2} \pm 0.1$ | $73.2 \pm 0.1$ |
| Cosine softmax | $\mathbf{65.8} \pm 0.1$ | $4.6 \pm 0.1$ | $\mathbf{24.8} \pm 0.1$ | $38.7 \pm 0.1$ | $\mathbf{72.5} \pm 0.1$ |
| Sigmoid | $\mathbf{65.9} \pm 0.1$ | $3.3 \pm 0.0$ | $22.6 \pm 0.1$ | $36.6 \pm 0.1$ | $74.6 \pm 0.1$ |

**Table B.3: Regularizers and alternative losses may or may not improve calibration.** We report negative log likelihood (NLL) and expected calibration error (ECE) for each loss on the ImageNet validation set, before and after scaling the temperature of the probability of the distribution to minimize NLL, as in Guo et al. (2017). ECE is computed with 15 evenly spaced bins. For networks trained with sigmoid loss, we normalize the probability distribution by summing probabilities over all classes.

| Loss/regularizer | Uncalibrated | | With temperature scaling | |
|---|---|---|---|---|
| | NLL | ECE | NLL | ECE |
| Softmax | $0.981 \pm 0.002$ | $0.073 \pm 0.0001$ | $0.917 \pm 0.002$ | $0.027 \pm 0.0004$ |
| Dropout | $0.971 \pm 0.002$ | $0.074 \pm 0.0009$ | $0.905 \pm 0.002$ | $0.031 \pm 0.0002$ |
| Label smoothing | $0.947 \pm 0.001$ | $\mathbf{0.016} \pm 0.0007$ | $0.941 \pm 0.001$ | $0.044 \pm 0.0004$ |
| Extra final layer $L^2$ | $0.976 \pm 0.002$ | $0.081 \pm 0.0003$ | $0.908 \pm 0.002$ | $0.038 \pm 0.0006$ |
| Logit penalty | $1.041 \pm 0.001$ | $0.090 \pm 0.0003$ | $0.995 \pm 0.001$ | $0.055 \pm 0.0004$ |
| Logit normalization | $0.965 \pm 0.001$ | $0.069 \pm 0.0002$ | $0.949 \pm 0.001$ | $0.049 \pm 0.0003$ |
| Cosine softmax | $\mathbf{0.912} \pm 0.002$ | $0.066 \pm 0.0006$ | $\mathbf{0.895} \pm 0.002$ | $0.043 \pm 0.0008$ |
| Sigmoid | $0.944 \pm 0.002$ | $0.044 \pm 0.0003$ | $0.914 \pm 0.002$ | $\mathbf{0.019} \pm 0.0002$ |

**Table B.4: Training accuracy of ResNet-50 models.**

| Loss/regularizer | Top-1 Acc. (%) | Top-5 Acc. (%) |
|---|---|---|
| Softmax | $93.61 \pm 0.01$ | $99.33 \pm 0.002$ |
| Squared error | $91.65 \pm 0.01$ | $98.59 \pm 0.002$ |
| Dropout | $92.25 \pm 0.01$ | $99.03 \pm 0.003$ |
| Label smoothing | $93.62 \pm 0.04$ | $99.43 \pm 0.007$ |
| Extra final layer $L^2$ | $91.62 \pm 0.01$ | $98.85 \pm 0.003$ |
| Logit penalty | $93.04 \pm 0.01$ | $99.13 \pm 0.002$ |
| Logit normalization | $92.86 \pm 0.01$ | $99.01 \pm 0.003$ |
| Cosine softmax | $92.47 \pm 0.01$ | $98.75 \pm 0.004$ |
| Sigmoid | $93.22 \pm 0.01$ | $99.19 \pm 0.002$ |

## C    RESULTS OF COMBINING REGULARIZERS/LOSSES

**Table C.1: Combining final-layer regularizers and/or improved losses does not enhance performance.** ImageNet holdout set accuracy of ResNet-50 models when combining losses and regularizers between models. All results reflect the maximum accuracy on the holdout set at any point during training, averaged across 3 training runs. Accuracy numbers are higher on the holdout set than the official ImageNet validation set. This difference in accuracy is likely due to a difference in image distributions between the ImageNet training and validation sets, as previously noted in Section C.3.1 of Recht et al. (2019).

|  | Baseline | Label smoothing $(\alpha = 0.1)$ | Sigmoid | Cosine softmax $(\tau = 0.05)$ |
|---|---|---|---|---|
| Baseline | 79.9 | 80.4 | 80.6 | 80.6 |
| Dropout $(\beta = 0.7)$ | 80.3 | 80.3 | 80.3 | 80.2 |
| Dropout $(\beta = 0.8)$ | 80.2 | 80.4 | 80.4 | 80.4 |
| Dropout $(\beta = 0.9)$ |  | 80.3 | 80.5 | 80.6 |
| Dropout $(\beta = 0.95)$ |  | 80.4 | 80.6 | 80.7 |
| Logit penalty $(\gamma = 5 \times 10^{-5})$ | 80.4 | 80.3 | 80.5 | 80.6 |
| Logit penalty $(\gamma = 1 \times 10^{-4})$ | 80.4 | 80.3 | 80.5 | 80.5 |
| Logit penalty $(\gamma = 2 \times 10^{-4})$ | 80.4 | 80.3 | 80.4 | 80.5 |
| Logit penalty $(\gamma = 4 \times 10^{-4})$ | 80.4 | 80.2 | 80.3 | 80.5 |
| Logit penalty $(\gamma = 6 \times 10^{-4})$ | 80.5 | 80.2 | 80.3 | 80.5 |
| Logit normalization $(\tau = 0.02)$ |  |  | 80.4 |  |
| Logit normalization $(\tau = 0.03)$ | 80.3 |  | 80.6 |  |
| Logit normalization $(\tau = 0.04)$ | 80.4 |  | 80.6 |  |
| Logit normalization $(\tau = 0.05)$ | 80.3 |  | 80.5 |  |
| Logit normalization $(\tau = 0.06)$ | 80.3 |  | 80.5 |  |
| Cosine normalization $(\tau = 0.045)$ | 80.6 |  | 80.5 |  |
| Cosine normalization $(\tau = 0.05)$ | 80.6 |  | 80.6 |  |
| Cosine normalization $(\tau = 0.06)$ | 80.4 |  | 75.3 |  |

**Table C.2: AutoAugment and Mixup provide consistent accuracy gains beyond well-tuned losses and regularizers.** Top-1 accuracy of ResNet-50 models trained with and without AutoAugment, averaged over 3 (with AutoAugment) or 8 (without AutoAugment) runs. Models trained with AutoAugment use the loss hyperparameters chosen for models trained without AutoAugment, but the point at which to stop training was chosen independently on our holdout set. For models trained with Mixup, the mixing parameter $\alpha$ is chosen from $[0.1, 0.2, 0.3, 0.4]$ on the holdout set. Best results in each column, as well as results insignificantly different from the best ($p > 0.05$, t-test), are bold-faced.

|  | Standard Augmentation | | AutoAugment | | Mixup | |
|---|---|---|---|---|---|---|
| Loss/regularizer | Top-1 (%) | Top-5 (%) | Top-1 (%) | Top-5 (%) | Top-1 (%) | Top-5 (%) |
| Softmax | $77.0 \pm 0.06$ | $93.40 \pm 0.02$ | $77.7 \pm 0.05$ | $93.74 \pm 0.05$ | $78.0 \pm 0.05$ | $93.98 \pm 0.03$ |
| Sigmoid | $\mathbf{77.9} \pm 0.05$ | $93.50 \pm 0.02$ | $\mathbf{78.5} \pm 0.04$ | $93.82 \pm 0.02$ | $\mathbf{78.5} \pm 0.07$ | $93.94 \pm 0.04$ |
| Logit penalty | $77.7 \pm 0.02$ | $\mathbf{93.83} \pm 0.02$ | $\mathbf{78.3} \pm 0.05$ | $\mathbf{94.10} \pm 0.03$ | $78.0 \pm 0.05$ | $93.95 \pm 0.05$ |
| Cosine softmax | $\mathbf{77.9} \pm 0.02$ | $\mathbf{93.86} \pm 0.01$ | $\mathbf{78.3} \pm 0.02$ | $\mathbf{94.12} \pm 0.04$ | $\mathbf{78.4} \pm 0.04$ | $\mathbf{94.14} \pm 0.02$ |

**Table C.3: Comparison with state-of-the-art.** All results are for ResNet-50 models trained with AutoAugment. Loss hyperparameters are the same as in Table C.2, but the learning schedule decays exponentially at a rate of 0.985 per epoch, rather than 0.975 per epoch. This learning rate schedule takes approximately $2\times$ as many epochs before it reaches peak accuracy, and provides a $\sim$0.4% improvement in top-1 accuracy across settings.

| Loss | Epochs | Top-1 (%) | Top-5 (%) |
|---|---|---|---|
| Softmax (Cubuk et al., 2019) | 270 | 77.6 | 93.8 |
| Supervised contrastive (Khosla et al., 2020) | 700 | **78.8** | 93.9 |
| *Ours:* | | | |
| Softmax | 306 | $77.9 \pm 0.02$ | $93.77 \pm 0.03$ |
| Sigmoid | 324 | $\mathbf{78.9} \pm 0.04$ | $93.96 \pm 0.06$ |
| Logit penalty | 346 | $\mathbf{78.6} \pm 0.07$ | $\mathbf{94.30} \pm 0.01$ |
| Cosine softmax | 308 | $\mathbf{78.7} \pm 0.04$ | $\mathbf{94.24} \pm 0.02$ |
| *Ours (with Mixup):* | | | |
| Sigmoid | 384 | $\mathbf{79.1} \pm 0.06$ | $94.28 \pm 0.03$ |
| Cosine softmax | 348 | $\mathbf{79.1} \pm 0.09$ | $\mathbf{94.49} \pm 0.01$ |

# D    SIMILARITY OF MODEL PREDICTIONS

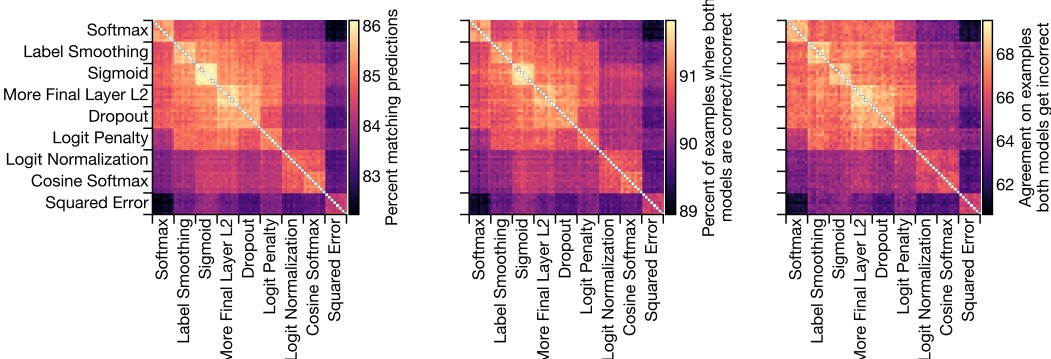

**Figure D.1: Different ways of measuring similarity of single-model ResNet-50 predictions yield similar qualitative results.** See also Figure 1.

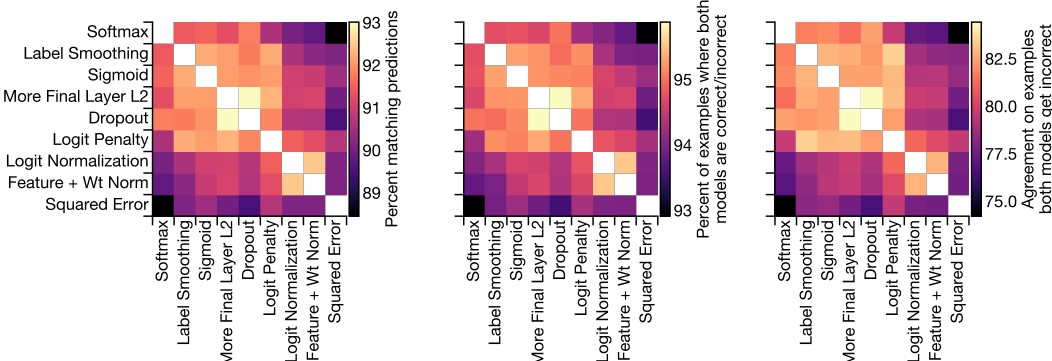

**Figure D.2: Ensemble predictions are substantially more similar than single-model predictions.** Predictions of the ensemble were computed by taking 8 ResNet-50 models trained from different random initializations with the same loss and picking the most common top-1 prediction for each example.

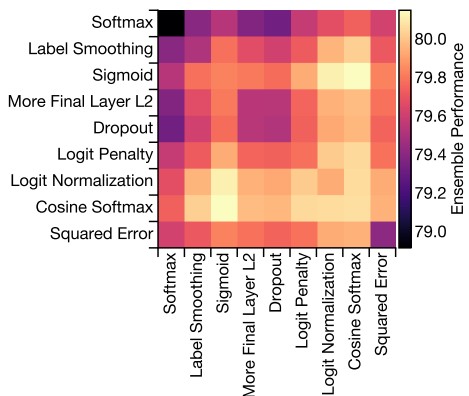

**Figure D.3: Ensembling models trained with different losses provides only modest performance benefits.** Ensembles consist of 8 ResNet-50 models, half of which are trained with the objective on the x-axis, the other half with the objective on the y-axis. The ensemble prediction is the modal class prediction of the 8 models.

# E  OTHER CLASS SEPARATION METRICS

**Table E.1: Comparison of class separation under different distance metrics.** Cosine (mean-subtracted) subtracts the mean of the activations before computing the cosine distance. All results reported for ResNet-50 on the ImageNet training set.

| Loss/regularizer | Cosine | Cosine (mean-subtracted) | Euclidean distance |
|---|---|---|---|
| Softmax | $0.3494 \pm 0.0002$ | $0.3472 \pm 0.0002$ | $0.3366 \pm 0.0002$ |
| Squared error | $0.8452 \pm 0.0002$ | $0.8450 \pm 0.0002$ | $0.8421 \pm 0.0007$ |
| Dropout | $0.4606 \pm 0.0003$ | $0.4559 \pm 0.0002$ | $0.4524 \pm 0.0003$ |
| Label smoothing | $0.4197 \pm 0.0003$ | $0.4124 \pm 0.0004$ | $0.3662 \pm 0.0005$ |
| Extra final layer $L^2$ | $0.5718 \pm 0.0006$ | $0.5629 \pm 0.0005$ | $0.5561 \pm 0.0005$ |
| Logit penalty | $0.6012 \pm 0.0004$ | $0.5950 \pm 0.0004$ | $0.5672 \pm 0.0004$ |
| Logit normalization | $0.5167 \pm 0.0002$ | $0.5157 \pm 0.0002$ | $0.5326 \pm 0.0002$ |
| Cosine softmax | $0.6406 \pm 0.0003$ | $0.6389 \pm 0.0003$ | $0.6406 \pm 0.0003$ |
| Sigmoid | $0.4267 \pm 0.0003$ | $0.4315 \pm 0.0003$ | $0.4272 \pm 0.0003$ |

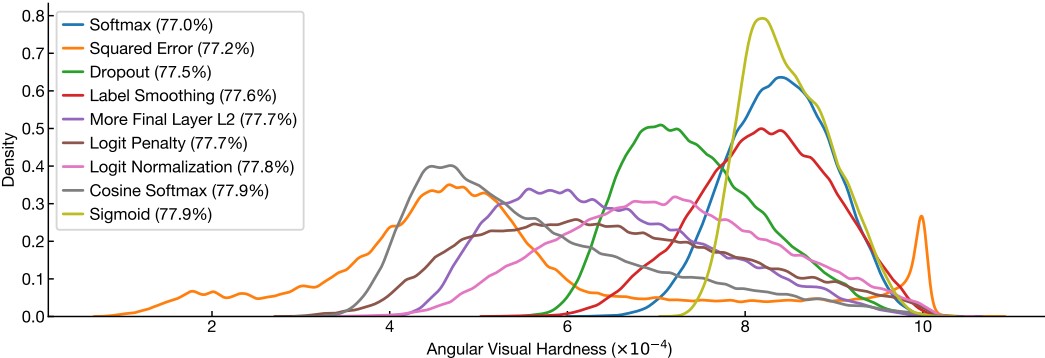

**Figure E.1: Angular visual hardness of different loss functions.** Kernel density estimate of the angular visual hardness (Chen et al., 2019) scores of the 50,000 examples in the ImageNet validation set, computed with a Gaussian kernel of bandwidth $5 \times 10^{-6}$, for ResNet-50 networks trained with different losses. Legend shows ImageNet top-1 accuracy for each loss function in parentheses. Although alternative loss functions generally reduce angular visual hardness vs. softmax loss, sigmoid loss does not, yet it is tied for the highest accuracy of any loss function.

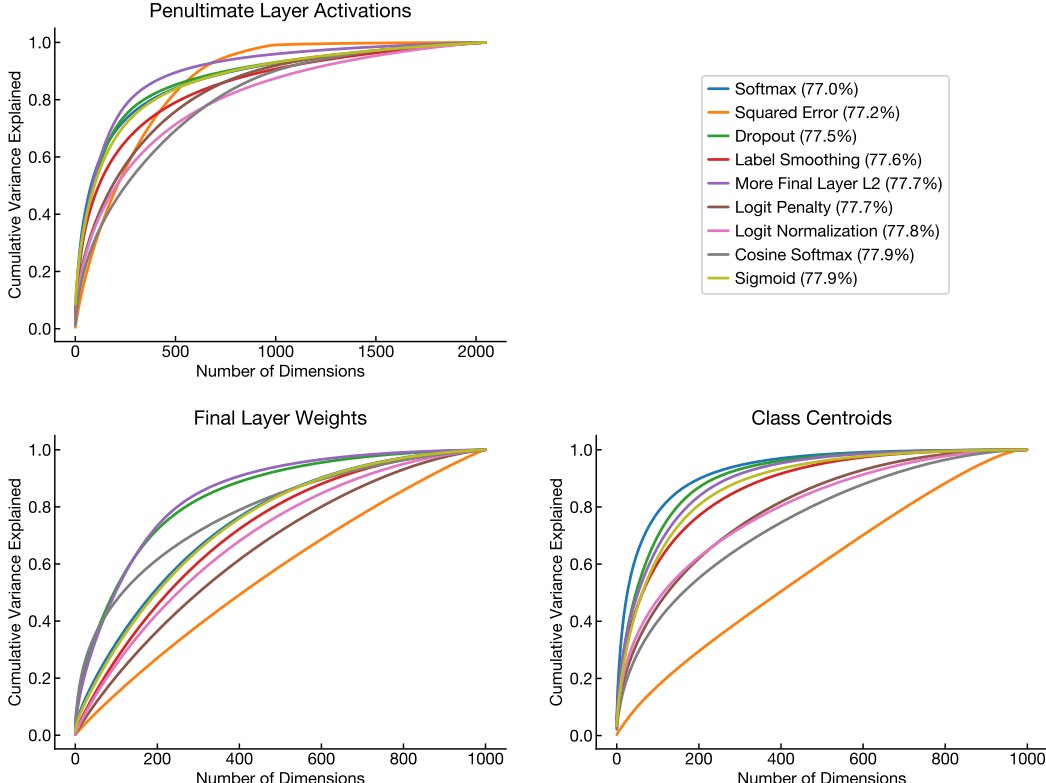

**Figure E.2: Singular value spectra of activations and weights learned by different losses.** Singular value spectra computed for penultimate layer activations, final layer weights, and class centroids of ResNet-50 models on the ImageNet training set. Penultimate layer activations and final layer weights fail to differentiate sigmoid cross-entropy from softmax cross-entropy. By contrast, the singular value spectrum of the class centroids clearly distinguishes these losses.

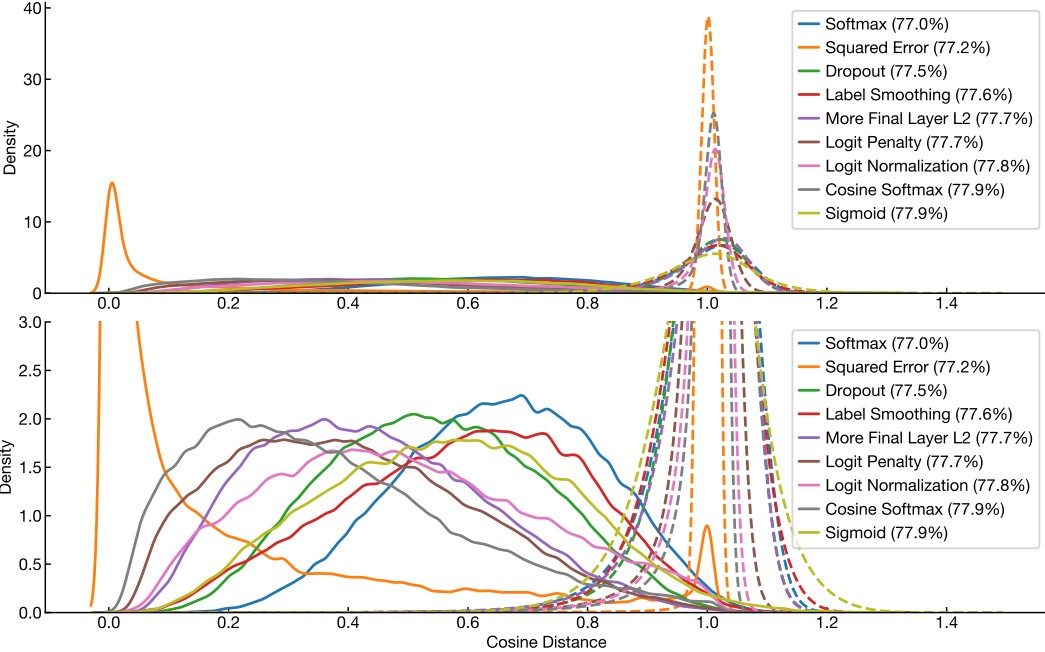

**Figure E.3: The distribution of cosine distance between examples.** Kernel density estimate of the cosine distance between examples of the same class (solid lines) and of different classes (dashed lines), for penultimate layer embeddings of 10,000 training set examples from ResNet-50 on ImageNet. Top and bottom plots show the same data with different y scales.

