# OpenReview forum: "Demystifying Loss Functions for Classification"
_ICLR.cc/2021/Conference — Reject_

### Official Review · AnonReviewer1 · 2020-10-29
**An investigation paper of existed loss functions. Seems no surprising results.**

**Rating:** 5
**Confidence:** 3

**Review:**

The authors give a pretty thorough investigation of several existed losses of training deep networks in the task of supervised classification. The differences objectives are compared in terms of accuracy, calibration, out-of-distribution robustness, and predictions.
Several discoveries are presented from the experiments:
1) Regularizers and alternative losses did improve over vanilla softmax loss. But the effect of improvement is not additive.
2) Different objectives mainly affect the last few layers while the layers not close to the output seems similar.
3) All objectives that improve over vanilla softmax loss produce greater class separation in the penultimate layer of the network.

Pros:
+ A comprehensive analysis of different behaviors of different objectives.
+ The experiments setup seems fair.

Cons:
- The authors claimed
'''Our goal is instead to understand when one might want to use one loss function or regularizer over another and, more broadly, to understand the extent to which neural network performance and representations can be manipulated through the choice of objective alone.'''
I wonder if the study in this paper realizes the goal that the authors claimed. It seems it is hard to make a conclusion to decide which single objective performs better than others, even in the task of classification, with the existing methods and optimization methods. It seems to have little insight for future works that using novel methods or optimization frameworks.

I respect the efforts that the authors made to give a thorough analysis of existing common-used objectives. But since little novel technical contribution is made and the results seem not surprising, I am not sure whether such work is above the threshold of ICLR.

---

> ### Author Response · Authors · 2020-11-20
> **Response to AnonReviewer1**
>
> We thank the reviewer for their comments.
>
> > “It seems it is hard to make a conclusion to decide which single objective performs better than others…It seems to have little insight for future works that using novel methods or optimization frameworks”
>
> We agree that, in Section 3.1, there is no single loss function that performs better than all others on all benchmarks. However, for specific benchmarks (of calibration, robustness, and linear transfer), some loss functions are clear winners. Conditioning publication of a paper on the demonstration that a method performs uniformly better than others leads to papers that compare only against methods to which the method is superior, on benchmarks where the method is superior. We agree that this leads to more exciting papers, but it is not always what’s best for the field.
>
> We believe it will be useful for future work to have accuracy numbers obtained from a wide range of well-tuned baselines for their novel methods or optimization frameworks, if only to confirm that these novel methods and optimization frameworks yield real improvements. For example, as we show in Table C.3, although the recently proposed Supervised Contrastive Learning (Khosla et. al, NeurIPS 2020) method substantially outperforms a cross-entropy baseline, it does not outperform the other objectives we examine.
>
> > “But since little novel technical contribution is made and the results seem not surprising”
>
> Our main goal was to probe representational properties of neural networks trained with different loss functions, not to propose a novel technical method. What is “surprising” seems subjective and we are not entirely sure how we can address the reviewer’s concerns. We believe our findings, especially those in Section 3.3 and 3.4 regarding the relationships between loss functions and neural network hidden representations, provide novel and potentially useful insights that have not been explored in previous literature.

---

### Official Review · AnonReviewer2 · 2020-10-29
**review of "DEMYSTIFYING LOSS FUNCTIONS FOR CLASSIFICATION"**

**Rating:** 3
**Confidence:** 5

**Review:**

The paper attempts to analyze the softmax cross entropy loss and discuss useful properties it has in an intuitive and accessible way. The paper mainly presents comparisons between this loss function and two other (very standard as well) loss functions that are the "sigmoid cross entropy" and the "squared error" loss functions.

It is very interesting to read the paper and realize that the topics discussed here have been written in well cited and well established statistics and ML books for decades. The softmax cross entropy loss is the multinomial regression loss while the sigmoid cross entropy loss is no other than the standard binary  logistic regression loss and finally the "squared error" is the least squared loss.

Authors compare the performance of these three on a few benchmarks.

I think it's good to read the "Elements of Statistical Learning" and get a clear exposition to what science has been up to for decades before getting too deep into deep learning.

---

> ### Author Response · Authors · 2020-11-20
> **Response to AnonReviewer2**
>
> This review is condescending and has no concrete content. Given that our paper focuses on empirical evaluation of neural networks and analysis of neural network hidden representations, we do not see which of our results the reviewer believes could have been derived from “well established statistics and ML books.”
>
> > “It is very interesting to read the paper and realize that the topics discussed here have been written in well cited and well established statistics and ML books for decades….softmax cross entropy loss is the multinomial regression loss...”
>
> In case it was unclear from the context, we do not claim any novelty for the loss functions described in Section 2; our paper’s contribution is the study in Section 3. We are thus puzzled by the reviewer’s statements relating different loss functions to losses used in generalized linear models. We believe these relationships are common knowledge, but we’re not entirely sure what knowledge the reviewer thinks can be gleaned from them.
>
> Even for linear models, in the absence of knowledge of the true data distribution, the choice of loss function is typically empirical, hence the widespread usage of both SVMs and logistic regression for classification problems. Section 3.1 and 3.2 are empirical studies of the performance and predictions of neural networks trained on images, where the data distribution is clearly analytically intractable, but likely similar across different image datasets. Theory of linear models is also clearly inadequate to explain the properties of neural network hidden representations, to which we dedicate Sections 3.3 and 3.4 of the paper.
>
> If the reviewer thinks there is a section of Elements of Statistical Learning from which the results in Section 3 of our paper can be derived, we challenge the reviewer to point to a specific chapter and outline the derivation.
>
> > “I think it's good to read the "Elements of Statistical Learning" and get a clear exposition to what science has been up to for decades before getting too deep into deep learning.”
>
> Although we accept that our paper has little to offer to a reviewer who is not interested in “getting too deep into deep learning,” we are puzzled to receive this comment given that the paper was submitted to the International Conference on Learning Representations.

---

### Official Review · AnonReviewer3 · 2020-10-29
**Very interesting work on practical aspects for training DNN models.**

**Rating:** 6
**Confidence:** 4

**Review:**

The paper presents a study on different loss functions and regularizers that apply on the penultimate layer and show how the choice of the loss function/regularizer can affect the performance of classifiers. The authors experiment with image datasets and study multiple parameters and side effects of the loss functions/regularizers.

This is an interesting paper with multiple experiments. It is a practical paper and provides multiple results on experiments for image classification. It is easy to follow and flows well.

Some weak points are the following;

- I find the datasets a bit limited and I think it would be nice if the authors could add extra datasets that go also beyond image classification.
- In the same spirit what it misses is an extension on a large-scale scenario. Many practical applications have very large output space. Experiments on such scenarios would help to better understand benefits/limitations of the different loss functions. Also, the authors could extend to other objectives like negative sampling etc.
- Some analysis on the individual classes is missing also. So, what do you observe when you have imbalanced classes?

As most of the impact is on the penultimate layers would it make sense to have a couple of experiments where you tune a model on another task. If I am not mistaken you trained from scratch.  It would be nice for example to have some results from another domain also like an NLP task.

Overall I think is a good paper.

---

> ### Author Response · Authors · 2020-11-20
> **Response to AnonReviewer3**
>
> We appreciate the reviewer’s positive feedback. We agree that investigations of datasets beyond image classification, on applications with large output spaces, on imbalanced datasets, and in transfer settings are important, and these are areas where in-depth evaluation and analysis would be useful for the research community. That said, we had to constrain the scope of the study to make a rigorous investigation feasible within a reasonable time frame and a finite computation budget, and we decided to focus our effort on popular computer vision datasets in this paper. Our analysis serves as a first step toward understanding the relationship between representations learned and loss functions used.

---

### Official Review · AnonReviewer4 · 2020-10-29
**Well written paper with comprehensive experiments, but lacking novelty of results**

**Rating:** 4
**Confidence:** 4

**Review:**

The paper benchmarks various  training objectives, including loss functions and regularization schemes and measures accuracy, calibration and out-of-distribution robustness. The paper also studies how these objectives affect representations in various layers of the network.


Pros
+ Paper is well written, and the narrative is coherent.
+ Informative plots for the most part.
+ Numerous experiments.

Cons
- Unfortunately, novelty is very limited and this is mostly a restating/confirmation of results of earlier work.
- Examples:  Sec 3.1 "Regularizers and alternate losses increase accuracy" -- what is new here? We already know this, otherwise these alternatives to vanilla cross-entropy wouldn't exist.
- Sec 3.2 "Different losses produce different predictions" : again, why is this surprising? If the losses produced the same predictions, then they would all have the same accuracy. And we know this is not the case.
- "Mixup and auto-augment improve performance " : already known.
- "label smoothing improves calibration" : already known

 The only possibly novel contribution is that losses behave similarly in the earlier layers, with most of the separation happening at the final layers. But this line --i.e., entanglement or separation behaving differently  in earlier vs. later blocks -- was already explored in Frosst et al 2019, so again here the novelty impact is limited.


minor
- dendrogram isn’t very intuitive. can the authors show a 2-d cluster representation of the model predictions (for all seeds, and for all losses?)

Suggestions on improving novelty:
- For a conference devoted to learning representations, the authors should investigated  into why, for example,  different random seeds of squared error loss disagree with each other so much, while some other losses are more self-consistent.

- Different losses produce different predictions. but given that top-1 accuracies are nearly the same (table 1), the difference in figure 2 must  mostly be contained in the predictions where the classifiers were wrong (respect to ground truth). in other words, there shouldn’t be a significant difference in the cases where the classifiers agree with ground truth -- can the authors look into this?

- At the start of the paper it is stated that  "our goal is instead to understand when one might want to use
one loss function or regularizer over another" : however, I do not see the paper providing any additional insights into this beyond what is already known.

In summary, lot of space is devoted to confirming earlier results, and while the  benchmarking is appreciated, a paper purely devoted to benchmarking but that only ends up confirming earlier results does not meet the bar.


References:

+ Frosst et al .Analyzing and Improving Representations with the Soft Nearest Neighbor Loss, ICML 2019

---

> ### Author Response · Authors · 2020-11-20
> **Response to AnonReviewer4 (part 1)**
>
> We thank the reviewer for their praise of our writing and plots. However, we challenge the reviewer’s claim that the novelty of our paper is limited. We address the reviewer’s concerns below.
>
> > Sec 3.1 "Regularizers and alternate losses increase accuracy" -- what is new here?
>
> As stated in our general response to all reviewers, we believe most of the paper’s novelty lies in Sections 3.3 and Section 3.4. That said, Section 3.1 also presents some novel findings:
> - We demonstrate that cosine softmax substantially improves both log-likelihood and out-of-distribution robustness over the other losses examined.
> - We show that ImageNet top-1 and top-5 accuracies are poorly correlated among different losses.
> - We demonstrate that combining multiple output regularization schemes doesn’t improve accuracy, even though output regularization and data augmentation schemes (e.g., AutoAugment and Mixup) are roughly additive.
> Do you know of a paper that shows concrete support for any of these findings?
>
> > Sec 3.2 "Different losses produce different predictions" : again, why is this surprising? If the losses produced the same predictions, then they would all have the same accuracy. And we know this is not the case. [...] Different losses produce different predictions. but given that top-1 accuracies are nearly the same (table 1), the difference in figure 2 must mostly be contained in the predictions where the classifiers were wrong (respect to ground truth). in other words, there shouldn’t be a significant difference in the cases where the classifiers agree with ground truth -- can the authors look into this?
>
> We believe that the questions the reviewer asks regarding Section 3.2 are already addressed in Appendix Figure D.1 of the submitted paper. The left panel of this figure replicates Figure 1a; the middle panel shows the proportion of examples on which a pair of models are both either correct or incorrect; and the right panel shows agreement only on examples that pairs of models both get wrong. Although there are differences between these plots, they all look qualitatively similar.
>
> As the reviewer intuits, the proportion of predictions on which models disagree must be at least as large as the difference in accuracy. However, as the reviewer later notes, the differences in accuracy are very small relative to the differences in predictions. Moreover, similarity in accuracy between losses does not correspond to similarity in predictions: sigmoid cross-entropy (77.9% top-1 accuracy) yields predictions more similar to softmax cross-entropy (77.0%) than to cosine softmax (77.9%). Appendix Figure D.1 (right) eliminates any possible confound with accuracy by considering agreement only on examples both models get incorrect.
>
> Clearly there cannot be a difference in top-1 predictions if both models agree on the ground truth. But even when trained with the same loss function, different initializations result in substantially different sets of correctly classified examples, as shown in Appendix Figure D.1 (middle).
>
> > The only possibly novel contribution is that losses behave similarly in the earlier layers...entanglement or separation behaving differently in earlier vs. later blocks -- was already explored in Frosst et al 2019, so again here the novelty impact is limited
>
> Our understanding is that the reviewer believes that, since Frosst et al. have previously shown that class separation increases sharply at the end of the network, their results imply that the loss function does not affect the rest of the network. This does not seem quite right. Optimizing _a_ loss clearly affects all layers of the network; freezing the bottom half of the network would substantially reduce accuracy, and there are also large changes in representations of these layers from init. Because Frosst et al. investigate entanglement only in cross-entropy-trained networks, it is unclear to us how one could infer the effect of changing the loss on early layers from their results.
>
> In Section 3.3 of our paper, we show that changing the loss does not affect most of the layers. The investigation in this section uses centered kernel alignment (CKA), which measures similarity of activations across layers of networks trained from different random initializations, without reference to the classes to which examples belong.
>
> In Section 3.4 of our paper, we investigate class separation in networks’ penultimate layers. Our point in this section is to show that (1) better-performing loss functions seem to lead to higher class separation and (2) greater class separation is associated with less transferable features. We don’t believe either of these results are discussed by Frosst et al. Nonetheless, we will add Frosst et al. to the list of previous papers investigating class separation in our related work section.

---

> > ### Author Response · Authors · 2020-11-20
> > **Response to AnonReviewer4 (part 2)**
> >
> > > ...the authors should investigated into why, for example, different random seeds of squared error loss disagree with each other so much…
> >
> > This can be an interesting but rather detailed topic for further research, which we leave to future work. We are pleasantly surprised by the reviewer’s interest in this topic given their earlier statement that the results in Section 3.2 are not surprising.
> >
> > > In summary, lot of space is devoted to confirming earlier results, and while the benchmarking is appreciated, a paper purely devoted to benchmarking but that only ends up confirming earlier results does not meet the bar.
> >
> > As mentioned in our response to all reviewers, only Section 3.1 is dedicated to benchmarking, and this section comprises <1 page of our paper.

---

### Author Response · Authors · 2020-11-20
**Response to all reviewers**

We thank all reviewers for their time. It appears that discussing benchmark results of different losses distracted from our key contributions. We believe the benchmark results are useful for both researchers and practitioners, but we agree that such evaluation is not the most novel part of this paper. Out of the 4.5 pages of the paper dedicated to results, <1 page is devoted to benchmarking. The majority of our results section describes characteristics of representations of neural networks trained with different loss functions, a topic that we do not believe has been investigated in previous literature. Specifically:

 * In Section 3.3, we show that representations in the first two-thirds of the blocks of ResNet-50 networks trained with different loss functions are indistinguishable; the choice of loss function affects only later layers.
 * In Section 3.4, we examine how losses differ in the structure they impose on the penultimate layer. Losses that result in greater class separation also result in higher accuracy, but their features are less transferable to other tasks.

It seems the reviewers have reviewed a different paper from the one that we have submitted. We would appreciate feedback on Sections 3.3 and 3.4, which include novel findings regarding similarities and differences in the representations of networks trained with different loss functions. We believe this kind of study is well-suited for the International Conference on Learning Representations.

---

### Decision · Program_Chairs · 2021-01-07
**Final Decision**

**Decision:**

Reject

**Comment:**

The paper presents an extensive empirical evaluation of several loss functions and regularization techniques used in deep networks. The authors conclude that the classical softmax is significantly outperformed by the other approaches, but there is no clear winner among them. Moreover, the authors have noticed two interesting facts, (1) the choice of loss function affects only upper layers of neural networks with the lower layers being very similar to each other, (2) losses that result in greater class separation also result in higher accuracy, but their features are less transferable to other tasks.

I agree with the authors that the comments of Reviewer 2 are shallow and not informative. Therefore, they were not taken into account in making the final decision and, as AC, I read the paper very carefully. Regarding Reviewer 4, however, I found his comments to be valid. There is a message that the authors want to communicate, and the reader that needs to decode this message using a noisy channel. Therefore, I encourage the authors to accordingly revise the paper to make the message much clearer.

The experimental papers that compare a wide spectrum of methods are always hard to judge and this judgement is often very subjective. There are several seminal papers of this type, but not so many for several reasons. I agree with the authors that such studies are very valuable and give evaluation being not biased by authors of a given method. They are also very time- and resource-consuming. But there should be a general consensus how such an experiment should be performed. The authors of the particular methods should also be able to give right feedback to make their methods to be run appropriately. Therefore, there exist several websites and initiatives that try to fulfill these requirements. As said above, any paper of this type will be judged very subjectively.

The discoveries made by authors should also be presented in a different way. One should start with a hypothesis that, for example, the lower layers are not affected by the loss function and then perform appropriate theoretical and empirical studies to verify the hypothesis. The same applies to the other discovery. In that way the message of the paper would be much clearer. I suppose that analysis of each of the discoveries deserves its own paper.